

# An effective survey method for studying volant species activity and behavior at tall structures

Brynn E. Huzzen[1], Amanda M. Hale[2] and Victoria J. Bennett[1]

[1] Department of Environmental Sciences, Texas Christian University, Fort Worth, TX, United States of America

[2] Department of Biology, Texas Christian University, Fort Worth, TX, United States of America

## ABSTRACT

The effects of anthropogenic modification of air space on wildlife, particularly volant species, is not fully understood. Thus, it is essential to understand wildlife-interactions with tall structures to implement effective mitigation strategies. Yet, we are currently lacking standard protocols for visual surveys of wildlife behavior at such heights. Our study sought to determine an effective, repeatable method using readily available night vision and thermal technology to survey wildlife at tall structures. Using bats as the taxonomic group of interest, we (1) created a key to identify bats and their behavior, (2) compared the effectiveness of 2 different technologies, and (3) assessed optimal equipment placement to visually capture bat activity and behavior in proximity to wind turbine towers. For the latter, we tested thermal cameras at four distances from the base of the tower. The results of our study revealed that thermal cameras captured ~34% more flying animals than night vision at a 2 m distance. However, due to the heat signature of the turbine towers themselves, it was challenging to identify behaviors and interactions that occurred in close proximity to the towers. In contrast, it was difficult to identify bats approaching the towers using night vision, yet we were able to clearly observe interactions with the towers themselves. With regards to equipment placement, we visually captured more bats with the thermal cameras placed 2 m from the tower base compared to farther distances. From our findings, we recommend that when using either thermal or night vision technology at tall structures, they be placed 2 m from the base to effectively observe interactions along the length of these structures. In addition, we further recommend that consideration be given to the use of these two technology types together to effectively conduct such surveys. If these survey techniques are incorporated into standard protocols, future surveys at a variety of tall structures are likely to become comparable and repeatable, thereby more effectively informing any mitigation strategies that may be required.

Corresponding author
Victoria J. Bennett, v.bennett@tcu.edu

## INTRODUCTION

Air space is important habitat for many volant species, providing foraging sites, mating opportunities, vantage points for predators, and access to resources (*Alerstam, 1979*;

*Avila-Flores & Fenton, 2005*; *Diehl, 2013*). Thus, any anthropogenic use of the air space can potentially influence wildlife beneficially or detrimentally. For example, one detrimental outcome would be that airplanes are estimated to strike over 25,000 birds annually in the United States (*Erickson, Gregory & Young Jr, 2005*; *Pfeiffer, Blackwell & DeVault, 2018*). Yet, we have modified the air space in many other ways as well. For decades, we have been installing tall anthropogenic structures (defined here as lattice or monopole towers >20 m in height) including electrical, radio, meteorological, satellite, and cell phone towers, along with power lines and skyscrapers. Studies have confirmed that such tall structures can disorient migratory birds (*Avery, Springer & Cassel, 1976*; *Gehring, Kerlinger & Manville, 2009*), reduce breeding success (*Dahl et al., 2012*), and are a source of bird and bat fatalities due to collisions (*Crawford & Wilson Baker, 1981*; *Timm, 1989*; *Longcore et al., 2012*; *Loss, Will & Marra, 2015*). Projections of annual mortality vary, but studies have estimated in the United States that 23,000,000 and 6,600,000 bird fatalities/yr occur at power lines and communication towers respectively, and 234,000 bird and between 200,000–800,000 bat fatalities/yr occur at wind turbines (*Allison et al. 2019*). Demographic studies predict that if these mortality rates continue, they could lead to population declines in certain species (eg., *De Lucas et al., 2012*; *Balotari-Chiebao et al., 2016*; *Frick et al., 2017*; *Rodhouse et al., 2019*). Yet, the threats caused by the modification of the air space are increasing with ongoing wind energy installations and other technological advances associated with urbanization, such as an increase in cell phone towers (*Lu, McElroy & Kiviluoma, 2009*; *Vasenev et al., 2018*). There is, therefore, a need to (1) better understand the impacts of air space modification on wildlife and (2) potentially implement mitigation strategies that can effectively alleviate such impacts.

While some strategies have already been developed and are now standard practice, such as avian-safe protections on electrical wires (*Avian Power Line Interaction Committee, 2006*), there are still opportunities to develop further mitigation at tall structures through continued research. For example, research is currently investigating the effectiveness of acoustic and visual deterrents intended to discourage birds and bats from approaching structures, such as wind turbines, communication towers, and skyscrapers (e.g., *Arnett et al., 2013*; *Swaddle & Ingrassia, 2017*; *Goller et al., 2018*; *Dwyer et al., 2019*). Another example of a preventative measure being explored is operational minimization strategies in which wind turbine blades are prevented from spinning at times when bats are most likely at risk of collision (e.g., *Arnett et al., 2011*; *Martin et al., 2017*; *Hayes et al., 2019*).

Although these approaches can be effective, there is still much room for improvement as we do not fully understand why and how different species are interacting with tall structures (*Wang, Wang & Smith, 2015*; *Bennett & Hale, 2018*; *Bernardino et al., 2018*). There are three broad hypotheses for bat interactions with tall structures: (1) interactions with tall structures are purely coincidental; (2) resources, such as shelter, mating opportunities, movement corridors, foraging sites, or water sources, are available in close proximity to the structures, thus increasing the probability of interaction; and (3) the structures themselves provide or are perceived to provide a resource, thereby attracting bats to them (*Cryan & Barclay, 2009*). Due to the high numbers of bat fatalities reported at wind turbines, there have been several studies to date that have focused on discerning which of these hypotheses

may explain why bats are coming into contact with these structures (e.g., *Bennett & Hale, 2014*; *Rydell et al., 2016*; *Bennett & Hale, 2018*; *Long, Flint & Lepper, 2011*). One such study, for example, by *Jameson & Willis (2014)* suggested that bats were attracted to tall anthropogenic structures during migration and found that the bat calls recorded at such structures were primarily composed of social calls indicative of mating behavior. Another study by *Bennett, Hale & Williams (2017)* found bat feces in turbine door slats, indicating bats were using wind turbines as night roosts. While a further 3 studies have examined the stomach contents of bat carcasses retrieved from beneath wind turbines (*Valdez & Cryan, 2013*; *Rydell et al., 2016*; *Foo et al., 2017*), finding that many of the stomachs were not only full (a sign of recent foraging), but also contained a similar composition of invertebrates to those on and around the wind turbines.

Typically, behavioral surveys involving direct observations provide an effective method of exploring wildlife interactions, but such surveys conducted at height have, until recently, been challenging. With the advancement of technology, it has become easier to conduct surveys using affordable, high-quality equipment like night vision and thermal cameras; and in the last 10 years a number of behavioral studies have been conducted on volant species at tall structures (e.g., *Long, Flint & Lepper, 2011*; *Mirzaei et al., 2012*; *McAlexander, 2013*; *Watson, Keren & Davies, 2018*). For example, from behavioral surveys conducted at wind turbines, *Cryan et al. (2014)* suggested that bat behavior around these structures was similar to behavior observed at tall trees where bats are searching for roosts, potential mates, and insect prey. However, the use of such technology in this field of study is still relatively new and there are no standardized protocols or even recommended guidelines available, making quantitative comparisons between different studies difficult. For example, while some studies have opted to use thermal cameras (e.g., *Cryan et al., 2014*), others have used night vision (*McAlexander, 2013*), and we currently do not know the extent to which these different technologies may vary in the number and quality of behavioral observations recorded.

To address this uncertainty, we conducted a study to determine best practice techniques for assessing the behavior of bats at tall structures. We (1) created a customized classification key to identify bats from other flying animals and characterize specific behaviors; (2) compared the effectiveness of currently available technologies (night vision and thermal cameras) for surveying bats; and (3) assessed optimal thermal camera placement in proximity to a tall structure. Based on the findings of our study, we aim to make recommendations on survey methods that can be widely implemented when investigating wildlife interactions with tall structures.

## METHODS

### Study site

We conducted our study at a utility-scale wind energy facility that has been operational since 2008 in north-central Texas, USA (N33°43′53.538″, W97°24′18.186″). This facility consists of 75 1.5-MW General Electric wind turbines comprising an 80 m tower, a 2 m nacelle, and 3 40 m blades (maximum height = 122 m). The 48 km$^2$ wind resource area

encompasses a matrix of cattle pastures, hayfields, cultivated fields, and scrub woodland. Surveys conducted from 2009 to 2013 identified 7 bat species at this site, including 6 species found in post-construction fatality monitoring surveys (eastern red (*Lasiurus borealis*), hoary (*Lasiurus cinereus*), silver-haired (*Lasionycteris noctivagans*), tri-colored (*Perimyotis subflavus*), evening (*Nycticeius humeralis*), and Mexican free-tailed (*Tadarida brasiliensis*) bats), and one additional species, the canyon bat (*Parastrellus herperus*), that was recorded in acoustic surveys (*Bennett & Hale, 2014*; *Lindsey, 2017*; *Bennett & Hale, 2018*). Furthermore, bats have been observed in close proximity to the wind turbine towers at this facility (*McAlexander, 2013*). We, therefore, deemed the facility to be an appropriate location to survey bat behavior and activity at wind turbine towers.

**Behavioral observation surveys**

In 2016 during the fall bat migratory period from July to mid-August (*Krauel & McCracken, 2013*; *Bennett & Hale, 2018*), we conducted a series of behavioral surveys to explore bat interactions with wind turbine towers. We elected to focus our surveys on the tower monopoles rather than the entire rotor swept zone, as current literature has suggested that the towers themselves provide resources for bats (*Long, Flint & Lepper, 2011*; *McAlexander, 2013*; *Jameson & Willis, 2014*; *Bennett, Hale & Williams, 2017*; *Foo et al., 2017*). For these surveys, six turbines were selected that had high levels of bat fatalities recorded in the aforementioned fatality monitoring surveys (*Bennett & Hale, 2014*; *Bennett & Hale, 2018*).

At 1–2 turbine sites per night, we conducted surveys to record bat activity in proximity to the turbine tower. For this, we investigated the effectiveness of 2 currently available technologies previously used to study volant species: night vision (*Warren et al., 2006*; *Fuller, Hammond & Tomasi, 2012*) and thermal cameras (*Blowers et al., 2015*; *Matzner, Cullinan & Duberstein, 2015*; *Hayman et al., 2017*). Night vision technology functions by taking all the available light, including infrared light, and amplifies it to allow the user to see in the dark. In contrast, thermal cameras use differences in heat energy emitted by all objects (thermal radiation) making them visually distinguishable. For example, as warm-bodied animals tend to have higher temperatures than the surrounding environment, they become visible even when they are in the dark or otherwise hidden from view. Using both technologies simultaneously, we then estimated the number of flying animals observed, determined which kinds of flying animals could be identified (e.g., bird, bat, moth, etc.), and categorized specific bat behaviors that were readily discernable with each technology type.

A night vision setup consisted of an ATN NVM14 night vision scope attached to Sony HDR-PJ790 video camcorder and placed on a Manfrotto MT055XPRO3 tripod, and 2 ATN Super Long Range Infrared Illuminator IR450 lights mounted atop VELBON EF tripods. For the thermal setup, we used an Axis Q1932-E 19MM thermal camera mounted on a Manfrotto MT055XPRO3 tripod, connected via an Ethernet cable and a Netgear ProSAFE 8-Port Fast Ethernet PoE Switch to an HP Compaq 8510w laptop with Axis Companion software (version 3.20.010, Axis Communications AB, Lund, Sweden), and powered by a 12 Volt 35 Amp automotive battery through a Cen-Tech Power Inverter.
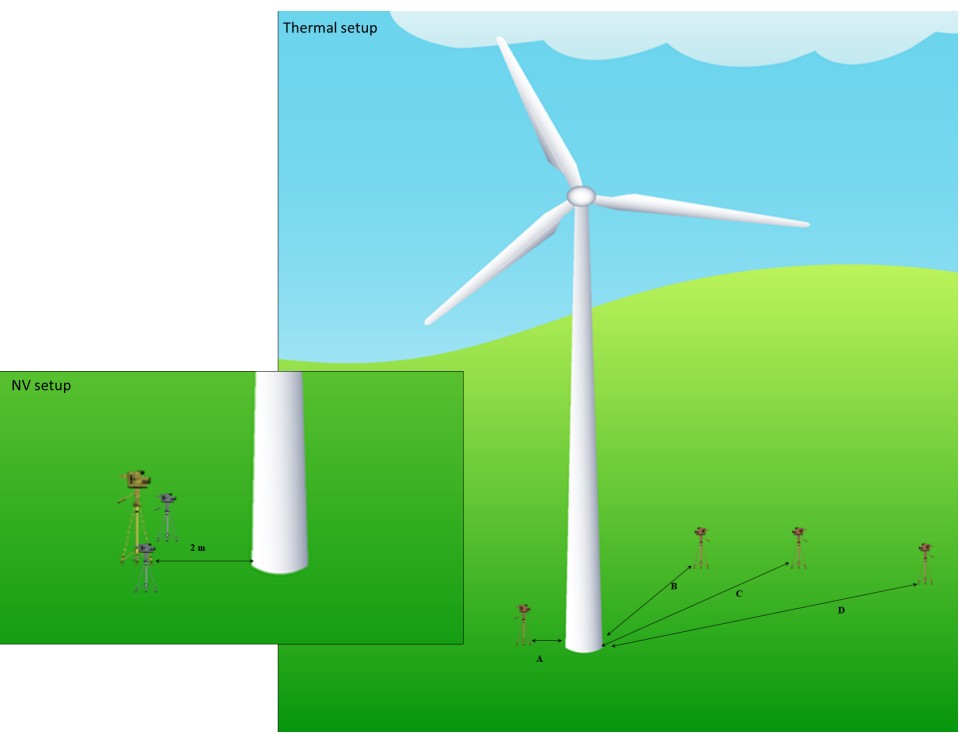

**Figure 1** **Diagram of the thermal and night vision setups used at our site.** Thermal cameras were tested at 4 distances (A) 2 m, (B) 25 m, (C) 50 m, and (D) 95 m from the wind turbine tower base, while night vision setups were kept at a distance of 2 m from the base and included 2 infrared lights to illuminate the tower.

From previous behavioral surveys using night vision at our site (2012 and 2013), we established that the optimal distance for identifying flying animals and their behavior using this technology was 2 m from the base of the tower (Fig. 1; *McAlexander, 2013*). Thus, as farther deployment distances were known to decrease detectability of flying animals, we opted to keep the night vision setups at this 2 m distance throughout this study. Furthermore, as bat activity can differ between the windward and leeward sides of wind turbines, we placed a night vision setup on either side of each survey tower (*Cryan et al., 2014*). We angled the field-of-view of both setups upward to capture the full length of the monopole (i.e., from ~10 m above the gravel pad to the lower surface of the nacelle hub ~80 m above ground level). We also placed the 2 infrared lights ~1 m from either side of each night vision setup, angling them upward to illuminate the tower surface.

For the thermal setups, there were currently no definitive recommendations regarding placement of equipment. Previous research using thermal cameras at wind turbines has varied camera placement from between 25 and 80 m from the base of the structure (*Horn, Arnett & Kunz, 2008*; *Cryan et al., 2014*). Thus, we sought to establish a thermal camera location that could effectively be used to survey the interactions of volant species with tall structures. For this, we incrementally tested a variety of distances to identify the camera location that yielded the highest number of observed flying animals per hour, and not only
optimized our ability to identify these animals (e.g., bird, bat, moth, etc.), but also allowed for specific behaviors to be distinguished.

We conducted a series of surveys with thermal cameras placed 2 m, 25 m, 50 m, and 95 m from the base of the leeward side of wind turbine towers (Fig. 1). We selected the leeward side as research has shown higher bat activity relative to the windward side of wind turbine towers (*Cryan et al., 2014*). For each distance tested, we adjusted the angle of the camera to maximize tower coverage from ∼10 m above the ground to the base of the nacelle (∼80 m above ground level) within the field-of-view. Note that the placement at 95 m from the base was tested because it captured the entire turbine, from the ground up to the top of the rotor swept zone, within the field-of-view.

During all surveys, we also used an ultrasonic acoustic detector to record species-specific bat activity around the towers. The acoustic recording equipment setup comprised an AR-125-EXT Ultrasonic Receiver and an iFR IV Integrated Field Recorder System from Binary Acoustic Technology, LLC with the microphone mounted atop a standard tripod. We pre-set the detectors to trigger at frequencies between 20 and 110 kHz at a gain threshold of 12.0 dB, trigger volume of 12.0 dB, and a duration of 4.0 s. Sound files were recorded as 4-second standard .wav files. Note that the detection range was limited to a maximum of 45 m (frequency-dependent). These detectors were placed at the base of each turbine alongside the night vision set-up on the leeward side of the turbine (Fig. 1). Ultrasonic detectors were turned on prior to starting the behavioral surveys and turned off when surveys were completed each night.

A survey night began 20 min after sunset and continued for up to 200 min to encompass the primary bat activity period (*Hayes, 1997*; *Baerwald & Barclay, 2011*; *McAlexander, 2013*). Within this time, we conducted a series of 12 10-min trials (this trial length was selected to ensure the equipment was working during the surveys and allowed us to process the recorded footage more efficiently) and ensured that all cameras were turned on and off in sync. We used visual cues (hand swipes) to synchronize the footage in the processing stage. Note that night vision scopes were not used for the first trial as they would not function during low light levels. Prior to each trial, we recorded the temperature (°C), wind speed (km/hr), and gust speed (km/hr), and did not conduct trials if temperatures were <5 °C, wind speeds were >24 km/hr, wind gusts were >32 km/hr, or if it was raining.

We processed all trial recordings using Studiocode video analysis software (version 5, Studiocode Business Group, Sydney, AU). In this software, we marked and timestamped any flying animals ≥8 cm (i.e., the length of the long axis of the smallest bat locally recorded, the tri-colored bat; *Ammerman, Hice & Schmidly, 2012*) that were observed within 2 m of the turbine tower. For the latter, we superimposed an 'observation zone' onto each video in Studiocode that was scaled using features, such as known flange and nacelle widths, from the wind turbine towers. By recording flying animals within this observation zone only, we kept the sampling area within the field-of-view consistent among the 4 different thermal camera distances we tested. We defined a 'bat' as any animal that had a silhouette resembling the body shape of a bat (i.e., visible head with ears, robust body, tapering wings serrated along the posterior edge, potentially with finger bones visible in the patagium; Figs. 2A–2B). We defined a 'non-bat' as any animal with characteristics that a bat would

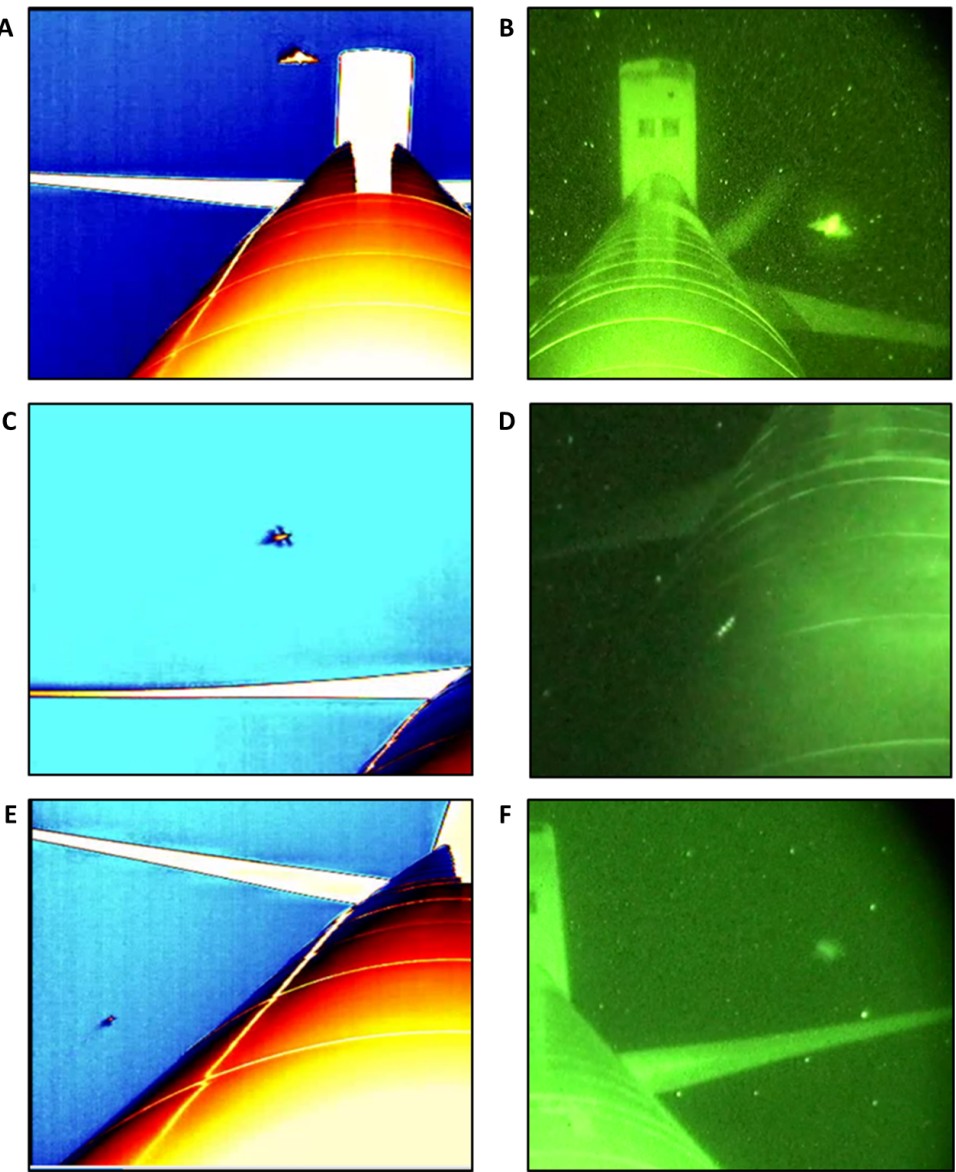

**Figure 2   Representative images from thermal cameras (A, C, E) and night vision (B, D, F) field-of-view showing the 3 categories of flying animals we observed at wind turbine towers.** (A–B) 'bats', (C–D) 'non-bats', and (E–F) 'undefined animals'.

not have (e.g., two pairs of wings, wings joined on the upper half of the body only, or thin bodies which are characteristic of some large insect species; presence of a beak, finger-like projections towards the ends of the wings, or long feathered tail which are characteristics of birds; Figs. 2C–2D). Lastly, we defined an unidentifiable animal as any animal that had no defining characteristics visible (Figs. 2E–2F). As the goal of our study was to maximize flying animal identification, an effective survey method would minimize the number of animals classified as 'undefined animals'.

To understand how bats were interacting with tall structures, one aspect of our survey method was to recognize specific behaviors. Thus, we defined the following 9 distinct bat behaviors: *passing*—when a bat flew across the field-of-view in a relatively straight flight path (≤1 turn); *reversing*—when a bat entered the field-of-view and turned back the way it came without passing the tower; *looping*—when a bat turned around at or after passing in front or behind the tower and returned back the way it came; *foraging*—when a bat flew in a zig-zag pattern with ≥2 changes in direction (i.e., turns); *chasing*—when a bat was closely followed by another bat; *skimming*—when a bat flew low over the tower, with its body parallel to the surface potentially making contact; *sweeping*—when a bat flew low over the tower and made contact with an outstretched wing tip; *colliding*—when a bat flew directly into the tower; and *gleaning*—when a bat hovered briefly over the surface of the tower before making contact with the surface (i.e., to potentially grab a prey item) before flying away. For the latter 4 behaviors, the reflections of bats in the turbine surface were used to confirm contact. We then classified the behaviors exhibited, where possible, by every 'bat' observed and indistinguishable behaviors (i.e., less than 1 s appearance in a corner of the field-of-view, or a lack of image clarity) were classified as *unknown*. As our study focused on the identification of bat behaviors in proximity to tower surfaces, we combined behaviors into 3 categories for the following analyses: *contact* (all behaviors in which a bat appeared to touch the turbine tower surface, including skimming, sweeping, colliding, and gleaning), *unknown*, and *all other behaviors* (including passing, reversing, looping, foraging, and chasing). As our survey goal was to facilitate behavioral identification, an effective method would minimize the number of *unknown* behaviors. Note that night vision and thermal camera recordings were scored separately to avoid bias and all footage was reviewed and scored independently by 2 separate individuals.

To determine whether flying animal detectability, identification, and behavioral classification differed between night vision and thermal cameras placed at a range of distances from the turbine tower base, we calculated the difference in the number of flying animals, the number of bats and non-bats, and the number of classified behaviors detected using night vision and thermal cameras for each turbine night ($n = 21$ turbine nights with ≥1 flying animal detected during the full 120-min survey period). For each response variable, we used a one-way ANOVA to compare the performance of night vision and thermal cameras. Due to unequal variances among distance categories for the thermal cameras, we pooled the locations as near (2 m from the turbine tower base: $n = 12$ turbine nights) and far (25, 50, and 95 m: $n = 9$ turbine nights) in the analyses. To evaluate the effect of distance from the turbine tower on the performance of thermal cameras, we used the number of observations (flying animals, bats and non-bats, and classified behaviors) as our response variable and distance as our explanatory variable ($n = 21$ turbine nights with ≥1 flying animal detected during the full 120-min survey period). Due to unequal variances among distances, we again pooled the far locations and used a Welch's $t$-test to compare the mean number of observations for thermal cameras placed near (2m from the turbine tower base: $n = 12$ turbine nights) and far (25, 50, and 95 m: $n = 9$ turbine nights) from the wind turbine towers. For all statistical analyses, we used Minitab software (version 18, Pennsylvania, USA) with $\alpha = 0.05$.

## RESULTS

From 1 July to 10 August 2016, we detected 551 flying animals in night vision and thermal camera surveys on 28 turbine nights (Table 1). On 7 of these turbine nights, not one flying animal was recorded. During the remaining survey nights, we recorded 194 bat acoustic calls, from which we identified: *Lasiurus borealis* ($n = 103$); *Lasiurus cinereus* ($n = 1$); *Lasionycteris noctivagans* ($n = 7$); *P. subflavus* ($n = 39$); and *N. humeralis* ($n = 44$).

In the first surveys we conducted, thermal cameras were placed 95 m from the tower base ($n = 12$ turbine nights with $\geq 1$ flying animal). At this distance, we found that flying animals were indistinguishable from the horizon because they were saturated by the infrared radiation signature it generated. In contrast, we noted that these flying animals remained visible in night vision recordings (setup located 2 m from tower base) and we were able to identify distinct bat behaviors, including close contact with the turbine tower. We then placed thermal cameras alongside the night vision setup 2 m from the base of the turbine tower and observed 50% more flying animals in the thermal camera footage compared to the night vision ($n = 12$ turbine nights with $\geq 1$ flying animal). Moreover, >200% more bats were identified in the thermal camera footage in comparison to the night vision (see Table 1). With regards to behavior, we found that >200% more behaviors were readily identified in thermal camera footage compared to the night vision and we were able to distinguish contact behaviors in the thermal camera footage (see Table 1).

When testing thermal cameras at 50 m and 25 m (in that order) from the turbine base, we opted to keep a second thermal camera 2 m from the base for an additional comparison as we knew that bats could be readily identifiable at this distance ($n = 2$ turbine nights for each distance). When thermal cameras were placed 50 m and 25 m from the tower base, we detected >60% fewer flying animals compared to either the thermal cameras or night vision at the 2 m distance (see Table 1). Moreover, >80% and >40% fewer bats were identified in the thermal cameras at 50 m and 25 m, respectively, compared to the thermal camera footage recorded at 2 m (see Table 1). Again, at the 2 m distance thermal cameras appeared to perform better than night vision during the surveys. Lastly, we were able to identify 80% fewer behaviors in thermal cameras at 2 m compared to 50 m, although identification was similar at 25 m and 2 m (see Table 1). Note that no contact behaviors were observed in any footage recorded during these surveys.

We found a significant difference in the number of flying animals detected between night vision and thermal cameras when the thermal cameras were placed near and far from the turbine tower base (Fig. 3A; $F_{1,19} = 33.07$, $P < 0.001$). With thermal cameras at 2 m from the tower base, we detected significantly fewer flying animals using night vision technology (95% CI [−9.6 to −3.5]). Yet with the thermal cameras at farther distances, we detected significantly more flying animals using night vision technology (95% CI [2.7–9.8]). For the number of bats and non-bats detected, we also found a significant difference between technology types (Fig. 3B; $F_{1,19} = 49.24$, $P < 0.001$). With thermal cameras at 2 m from the tower base, we detected significantly fewer bats and non-bats using night vision technology (95% CI: -10.2, -5.4), whereas at far distances we detected significantly more bats and non-bats with night vision technology (95% CI [1.8–7.3]). And finally, for the number

Huzzen et al. (2020), *PeerJ*, DOI 10.7717/peerj.8438

**Table 1 Number of flying animals, bats, and non-bats, and associated behaviors detected in surveys using night vision and thermal cameras from 1 July to 10 August 2016 (n = 28 turbine nights).** Shown in parentheses are the number of flying animals, bats, and non-bats, and associated behaviors detected in thermal cameras at 2 m distances. Totals are given in bold.

| Survey nights | # of turbines surveyed | Distance from turbine base | Thermal cameras | | | | | Night vision | | | | |
|---|---|---|---|---|---|---|---|---|---|---|---|---|
| | | | # flying animals | # bats | # non-bats | # behaviors identified | # contact behaviors | # flying animals | # bats | # non-bats | # behaviors identified | # contact behaviors |
| 6 | 2 | 2 m | 237 | 68 | 119 | 66 | 2 | 159 | 22 | 72 | 20 | 0 |
| 2 | 1 | 25 (2) m | 5 (13) | 3 (7) | 1 (4) | 3 (7) | 0 (0) | 12 | 5 | 3 | 5 | 0 |
| 2 | 1 | 50 (2) m | 14 (42) | 5 (25) | 0 (8) | 5 (25) | 0 (0) | 39 | 15 | 8 | 15 | 0 |
| 6 | 2 | 95 m | 3 | 2 | 0 | 2 | 0 | 27 | 19 | 2 | 19 | 1 |
| **16** | | | 259 (55) **314** | 78 (32) **110** | 120 (12) **132** | 76 (32) **108** | 2 (0) **2** | **237** | **61** | **85** | **59** | **1** |

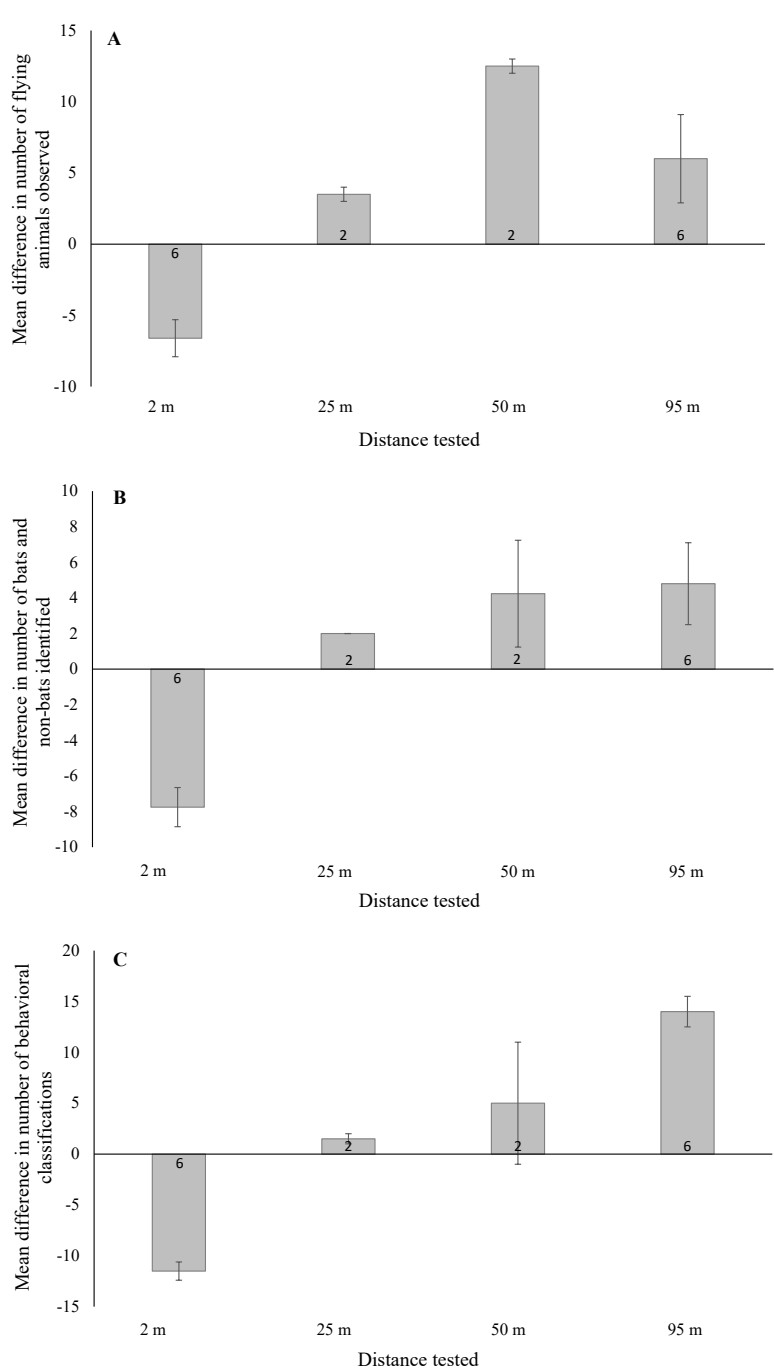

**Figure 3** **Mean ± SE difference in the total number of (A) flying animals (includes bats, non-bats, and undefined animals) detected, (B) bats and non-bats identified, and (C) behaviors categorized.** Differences were calculated from the total number of detections recorded using thermal cameras at 2 m, 25 m, 50 m, and 95 m from the wind turbine tower base minus the number of detections recorded by night vision technology at a distance of 2 m from the base. Numbers within the bars indicate the number of survey nights.

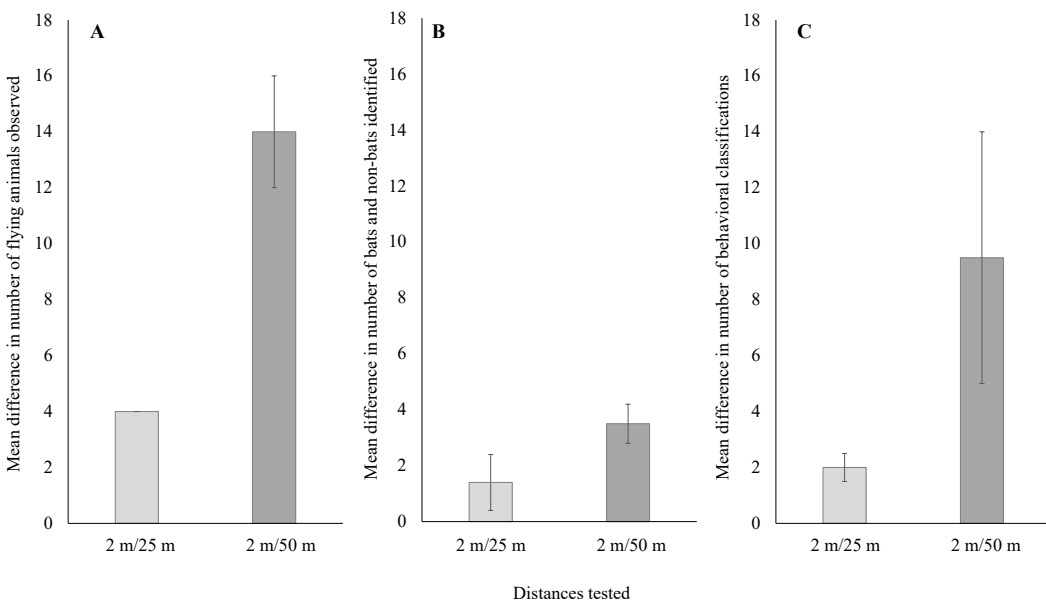

**Figure 4  Mean ± SE difference in (A) the total number of flying animals detected, (B) the number of bats and non-bats identified, and (C) the number of behaviors categorized.** Differences were calculated from the total number of detections recorded using thermal cameras at 25 m and 50 m from the wind turbine tower base minus the number of detections recorded by the thermal camera at a distance of 2 m from the base.

of classified behaviors, we also found a significant difference between technology types (Fig. 3C; $F_{1,19} = 26.89$, $P < 0.001$). With thermal cameras at 2 m from the tower base, we classified significantly fewer behaviors using night vision technology (95% CI [−5.6 to −2.0]), whereas at far distances we classified significantly more behaviors with night vision technology (95% CI [0.92–5.1]).

When comparing just thermal cameras, we also found a significant difference in performance that varied with distance from the turbine tower base. For the number of flying animals, detections using thermal cameras were significantly higher near turbine towers compared to farther away (Fig. 4A; Welch's $t$-test: $t_{14} = 6.05$, $P < 0.001$; 95% CI [11.1–23.3]). For the number of bats and non-bats, detections were also significantly higher near turbine towers compared to farther away (Fig. 4B; Welch's $t$-test: $t_{12} = 7.18$, $P < 0.001$; 95% CI [9.9–18.6]). Lastly, for the number of classified behaviors, these were also significantly higher near turbine towers compared to farther away (Fig. 4C; Welch's $t$-test: $t_{18} = 4.30$, $P < 0.001$; 95% CI [2.3–6.8]).

## DISCUSSION

Detectability, identification, and behavioral classification of flying animals varied between night vision and thermal camera recordings, and between thermal cameras positioned 2 m, 25 m, 50 m, and 95 m from the base of the wind turbine towers in our study. These findings indicate that behavioral observation surveys differing in their set-up could potentially yield different results. More specifically, our study revealed that data collected could vary due to

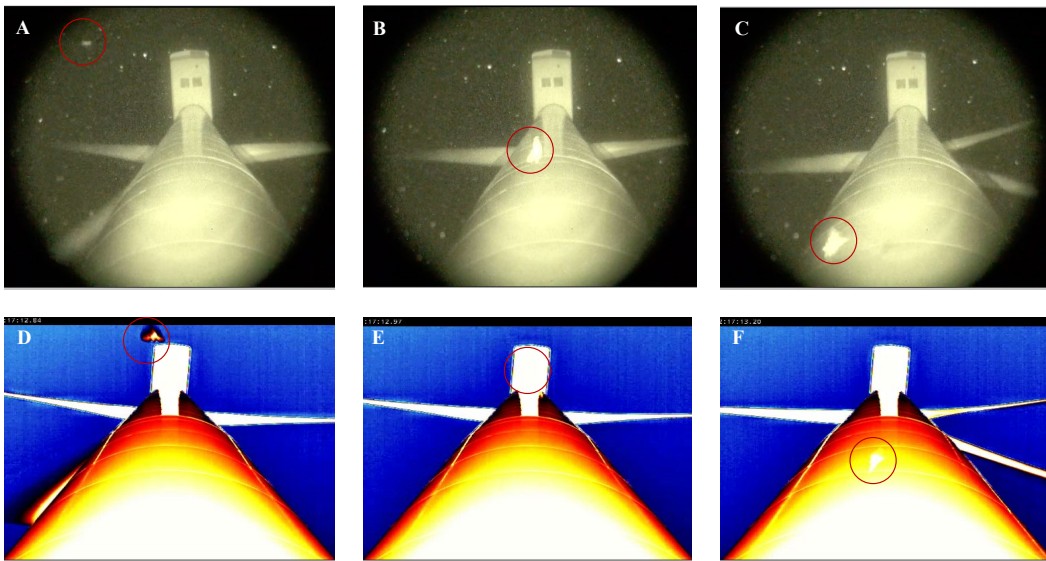

**Figure 5  Thermal and night vision images of bats (circled in red) approaching and passing closely in front of a wind turbine tower.** (A) shows that the bat is difficult to see when approaching the tower using night vision technology. (B–C) demonstrate that the bat is readily visible in front of the tower using night vision technology. In contrast, (D) shows that the bat is readily visible approaching the turbine tower using thermal cameras. (E–F) demonstrate that the bat becomes obscured by the infrared heat signature from the turbine tower.

limitations associated with the technology used. For example, flying animals and behaviors in front of tower surfaces were more readily identified using night vision technology; however, animals approaching the tower were virtually indistinguishable from the night sky (Fig. 5A). This limitation made it more challenging to detect flying animals approaching the towers and identify specific behaviors, unless the latter occurred directly in front of the tower (Figs. 5B–5C).

A similar detection limitation occurred with the thermal cameras, albeit to a lesser extent, as approaching animals could not be clearly seen in the lower half of the field-of-view at distances of 25 m, 50 m, and 95 m, due to an infrared radiation signature generated by the horizon. Moreover, while these distances provided a larger field-of-view in which flying animals approaching the wind turbine towers were observed earlier (potentially increasing detection), there were consequences in terms of resolution, subsequently reducing the identification of flying animals and behaviors effectively. For example, our ability to classify bat behavior in close proximity to the turbine towers was reduced at 25 m distances compared with 2 m distances, and it was difficult to detect approaching flying animals, let alone identify them, at the 95 m distance. At the 2 m distance, as the thermal cameras were angled up toward the wind turbine tower, there was no horizon in the field-of-view. Subsequently, approaching flying animals and behaviors were more visible and identifiable, even though the area in which approaching bats could be observed was limited to <1 m toward the base of the turbine tower to ~40 m out from the top of the tower monopole beneath the nacelle (Fig. 5D). Another drawback with the use of thermal cameras was that

the surfaces of the monopole and nacelle also generated an infrared radiation signature that tended to obscure visibility when animals passed directly in front of them (Figs. 5E–5F). Note that while it was not impossible to identify specific behaviors within these areas of the turbine, it did make scoring videos more challenging (reduced confidence) and time consuming.

In summary, our study revealed contrasting advantages and disadvantages in using both of the technologies tested. Ultimately, set-up selection depends on the objective of a study. If surveys are intended to record presence, activity patterns, and abundance of bats or other similar-sized flying animals, such as small birds, then thermal cameras at distances of 25 m from the turbine would maximize data collection. For example, comparing the night vision and thermal cameras observations of the flying animals, 34% were detected with thermal cameras and not with night vision technology, whereas only 1% were detected with night vision technology and not with thermal cameras. Moreover, at this distance both night vision and thermal cameras can be readily angled to incorporate different areas of the turbine in their field-of-view, whether it be the entire rotor swept zone or the majority of the turbine monopole and nacelle. Studies, such as *Roemer et al. (2017)*, that set out to determine rates of bat or bird collisions with wind turbines, whether bat activity can be used to predict fatality, or monitor the effectiveness of operational minimization strategies, would likely benefit from using thermal cameras at 25 m from the wind turbine base.

In comparison, if the goal of a study is to specifically explore why bats are coming into contact with wind turbines or other tall structures, then a set-up 2 m from the base of the structure would optimize the resolution needed to effectively conduct behavioral observation surveys. Studies, such as *Jameson & Willis (2014)*, that explore whether bats are attracted to tall structures for social interactions and foraging, mating, and roosting opportunities, could optimize data collection with the use of thermal cameras at a 2 m distance. Nevertheless, with the aforementioned limitations of both night vision and thermal cameras at 2 m, a study that either uses one or the other technology could lead to variations in animal identification and behavioral classification with the potential to bias the results of the study. For example, among the flying animals recorded in both the night vision and thermal cameras, 87% were identified as bats or non-bats with thermal cameras and 60% with night vision technology. Similarly, 66% of behaviors were classified with thermal cameras and 31% with night vision technology. Thus, our findings indicated that using the two technologies in combination with one another yields more reliable results, with thermal cameras optimizing our ability to detect approaching animals and night vision enhancing and making it much easier to identify animals and classify behavior at the wind turbine tower surfaces.

We also acknowledge that the shortcomings of thermal cameras in our study may have been exacerbated by high ambient temperatures at our site; other study locations with lower ambient temperature may not experience such radiation-related issues. Regardless, in areas with similar climates to our study site, we certainly recommend the combined technology setup. We also consider that these recommendations are relevant to any survey of flying animals conducted at tall metal structures.

Another consideration is the placement and number of night vision or thermal camera set-ups for the surveys. If economically feasible, we recommend that set-ups are positioned on more than one side of the structure to effectively detect flying animal presence and capture interactions around the entire structure. However, as previous studies at wind turbines have shown that the majority of activity occurs on the leeward side of the turbine (*Cryan et al., 2014*; *Hein & Schirmacher, 2016*), in these instances, one set-up may be effective. The challenge with using multiple set-ups with overlapping field-of-views is to avoid the duplication of observations. Viewing and scoring footage from multiple fields-of-view simultaneously would reduce the occurrence of this type of data replication happening.

In addition, for all surveys at wind turbines and other tall structures, we recommend the creation of an object identification key to allow comparable and repeatable flying animal identification. Note that our key was customized for our taxonomic group of interest, survey timing (e.g., day or night), and study site; we therefore recommend that keys are customized and validated for specific surveys. For example, the key created in our study could easily be modified for bird species that may be flying near wind turbines at night, and even used for the bigger-bodied invertebrates, such as moths, dragonflies, and grasshoppers.

Of course, as technology is quickly advancing and continues to improve with new models becoming commercially available each year, the issues highlighted in our study may become less problematic. Nevertheless, if we are to understand why, for example, bats are coming into contact with wind turbines, it is crucial to compare and collate data from different studies, past and present. By doing so, we can uncover those patterns in activity and bat behavior at regional, national, and even global scales, which in turn may help inform effective mitigation for reducing fatalities in wind energy facilities. Thus, it is important that we acknowledge the capabilities and limitations of the technology, such as resolution, used in each study, so that we can make appropriate comparisons.

## CONCLUSION

As the construction of anthropogenic structures continues to modify air spaces, behavioral surveys will only become more important (*Cousins et al., 2012*; *Arnett & Baerwald, 2013*; *Vasenev et al., 2018*). Furthermore, existing structures could be modified and improved (i.e., increased height of towers, length of wind turbine blades, etc.), thus understanding how these changes could impact wildlife should be considered (*Thomsen, 2009*). For these future studies, we have made recommendations to standardize surveys that would allow for comparison between different studies, structure types, and sites, thereby increasing our ability to effectively inform mitigation strategies to alleviate anthropogenic effects to wildlife.

## ACKNOWLEDGEMENTS

We express gratitude to all technicians who worked on this project: Ryan Conley, Martin McQueen, Cole Lindsey, and Matt Paulsen; and we would also like to thank the following

reviewers: Dr. Michael Slattery, Becky Johnson, Kathryn Smith, Martin McQueen, and Alyssa Austin.

### Funding

This work was supported by the U.S. Department of Energy, Office of Energy Efficiency and Renewable Energy –Wind & Water Power Program (DE-EE0007033) and TCU-NextEra Energy Resources Wind Research Initiative (P23113) with funding provided to Amanda M. Hale and Victoria J. Bennett. The funders had no role in study design, data collection and analysis, decision to publish, or preparation of the manuscript.

### Grant Disclosures

The following grant information was disclosed by the authors:
U.S. Department of Energy.
Office of Energy Efficiency and Renewable Energy –Wind & Water Power Program: DE-EE0007033.
TCU-NextEra Energy Resources Wind Research Initiative: P23113.

### Competing Interests

The authors declare there are no competing interests.

### Author Contributions

- Brynn E. Huzzen performed the experiments, analyzed the data, prepared figures and/or tables, authored or reviewed drafts of the paper, and approved the final draft.
- Amanda M. Hale conceived and designed the experiments, authored or reviewed drafts of the paper, and approved the final draft.
- Victoria J. Bennett conceived and designed the experiments, performed the experiments, analyzed the data, authored or reviewed drafts of the paper, and approved the final draft.

### Field Study Permissions

The following information was supplied relating to field study approvals (i.e., approving body and any reference numbers):

Access to the study site, Wolf Ridge Wind LLC, was granted by the site manager Carson Brazil on behalf of NextEra Energy Resources, LLC (NextEra Energy Resources, LLC owns and operates Wolf Ridge Wind LLC).

### Data Availability

The raw data is available in a Supplementary File.

### Supplemental Information

Supplemental information for this article can be found online at http://dx.doi.org/10.7717/peerj.8438#supplemental-information.

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
