# Peer review of "An effective survey method for studying volant species activity and behavior at tall structures"

_PeerJ, doi:10.7717/peerj.8438_

## Round 0.1 · original submission · Major Revisions

Both reviewers provide detailed questions, comments, and suggestions for improving the manuscript. In particular, they both suggest approaches for improving how the study is framed in the Introduction, clarifying the Methods, and enhancing the Discussion. I have selected “major revisions,” but I believe the revisions required are probably more “moderate” than “major”.

Reviewer 1 ·

Basic reporting

The paper satisfies most basic reporting requirements though it is unclear whether or not raw data will be shared.

Experimental design

See comments to author

Validity of the findings

See comments to author

Additional comments

The paper seeks to assess the effectiveness of two remote sensing methods to monitor flying animals in the vicinity of wind turbines, and based on results, suggest recommendations on their application. The concept has merit as continued growth is expected in wind energy development and methods of minimizing risk to flying animals are still under development.

The paper would benefit from matching the scope of its impact as described in the introduction to the actual data gathered. On lines 79-80, the paper purports to determine best practices for assessing behavior of flying animals at tall structures, but only one kind of structure is studied (turbines), one remote sensing technology is varied (thermal imaging), and one variable is examined (distance from the base of a turbine). Identifying “best practices” for the application of even a single technology would identify and explore a far greater list of issues including different sampling strategies and times of night (and times of the year), detection range for different sized objects and how those interact with background conditions (a concern described in the paper), the region of coverage (e.g., whether or not to image the rotor swept area), methods of quantification and statistical analysis, the competing benefits associated with different kinds of thermal imaging technology, and how all these may vary with conditions. (Regarding that second to last point, the usefulness of a given thermal imaging technology will be based on a variety of often competing factors including field of view, specific operating frequency/wavelength, cost, and dynamic range and sensitivity.) Likewise, lines 82-83 imply multiple technologies will be examined when only thermal imaging is varied; this is the heart of the paper and the intro and discussion should be structured around exploring the effects of thermal imaging flying animals at different ranges from turbines.

There are some concerns with methods, and the discussion in general is brief and could be much more exploratory. These are described in greater detail in the line-specific comments below.

Ln139. The field of view would increase with distance from the base, so an accurate comparison would require controlling for this difference such that the same field of view was being assessed across distances. The effect would be an even greater difference between sampling near the turbine v. further away. Best to correct in analyses, but at least mention this caveat in the discussion. Also, why was the rotor swept area not included in the field of view when that is the area impacting the vast majority of flying animals?

Ln159-160. Why not leave the recording equipment on throughout the 200 min time frame? How were time stamps synchronized across the recording devices?

Ln168. What is the significance of 8 cm? Does this refer to the long axis of biological targets?

Ln169. A bird has a head, body, and wings. Was the primary morphological difference patagium (bat) v. tail (bird)?

Ln171-172. Acoustic verification seems subject to considerable uncertainty. The acoustic source could be from a bat outside video coverage (unless these were highly directional mics, which seems unlikely). Moreover, video range presumably far exceeded the max range of acoustic monitoring (45 m), so only lower height targets stood the chance of being verified this way. And what was the max distance a bat-like target could be identified in thermal and NV data? It’s important to know given the role this equipment plays in determine risk to flying animals from rotors that begin some 40 m AGL. Likewise, the max resolvable height for insects is much lower, especially to resolve wings (Ln173).

Ln173. I cannot resolve a bird-like target in the figure.

Ln174. Like the concept of an ‘undefined’ or ‘unidentifiable’ category, but why call it a ‘possible bat’ if it could also be a possible bird or possible invertebrate? Leave as undefined or unidentified throughout.

Para beginning Ln177. How were these scored? One person? Did a second individual blindly score a subsample as a validation? Were NV and thermal imagery scored separately to avoid bias? BTW, avoid referring to thermal imagery/video/data as thermals, that term is unclear and has other meanings.

Ln185. Seems ‘sweeping’ would be challenging to identify from video….how do you know the wing touched?

Ln191-195. Like reduction to simpler categories.

Ln206. Use of KW test suggests data does not satisfy assumptions for a parametric test. These things should be stated.

Ln207. What is a Games Howell test and what is its specific purpose here?

Ln217. 146 trials in six nights is greater than the number of trials possible in six nights given the 12 10-min trials per night mentioned on Ln159.

Ln216+. Much of the next 30 lines or so could be placed in a table and made much easier to read and compare.

Ln239. See comment Ln217.

Ln244-245. Is this the point of saturation of the camera, or is it that the background temperature is indistinguishable from that of the target.

Ln250. “per hour”…but data were gathered in 10-min blocks?

Ln252-257. Mix of parametric and nonparametric among these four test results?

Ln275-278. So greater likelihood of uncertainty from data gathered closest to the target. Why would that be?

Ln289-290. The background is likely the same at 2 m as it is at other ranges (or nearly so). A combination of spreading loss over greater distance and limited dynamic range of the camera would cause weaker thermal signals from flying animals to merge with background.

Ln291+. What if data were gathered later in the evening giving the tower time to cool? Might bat-like targets be more detectable? Might a camera with more dynamic range (higher bit resolution) increase the likelihood of resolving subtle differences in temp between a bat and background be it the sky, cloud, or turbine?

Fig5 seems to offer little. Maybe let this result stand on the text alone.

·

Basic reporting

This study evaluates two approach to generating video imagery for studying bat behavior associated with wind turbines and other tall structures. The following are my comments on the manuscript. Thanks for letting me take an early look at this manuscript. I enjoyed reading it.

I would see the focus of the study as follows: (1) compare and contrast two approaches to generating video imagery of bats at tall structures; and (2) develop and use a taxonomy of behaviors for studying bat behavior associated with these structures. The current manuscript does a very good job of describing the technical methods used, but in my opinion, the discussion of behavior and placing the work in a broader context could be improved.

I don't quite feel like these authors provide a thorough review of the literature related to how bat researchers have tried to use video imagery to study bats at wind energy facilities, and I don’t feel like the authors thoroughly discuss how their approach fits into this long-term line of research. I also feel that the discussion is too short and doesn’t provide enough detail in comparing this work to the other research that has occurred, or consideration of how the behavioral results here relate to results observed by other researchers.

I would recommend that PeerJ consider publishing this manuscript after a revision.
Here are some specific comments for the author’s consideration:
- Line 43: Consider defining what is meant by a “tall” structure. How tall?
- Line 54: What specifically is meant by “population-level implications”?
- Line 66: Consider defining “deterrent”. Some readers will be unfamiliar with this term.
- Line 67: Likewise, consider defining “operational minimization”.
- Line 75: Consider citing the Cryan et al. (2014) paper here, along with the others.
- Line 83: Consider using “animals” instead of “objects”.
- Introduction, general: I would encourage a thorough review of the behavioral ecology literature related to flight behavior in bats. There is a rich literature here that goes largely un-cited.
- Introduction, general: consider restricting the purpose and focus of the paper to studying flying bats, then returning to the broader question of volant animals in the (longer) discussion.
- Line 95: consider including common names for each species.
- Lines 109-110: consider briefly describing the physics of these two approaches. For example, what wavelengths of light are they documenting, and any other important differences so readers who are unfamiliar with these technologies have a better sense for how they compare to each other.

Experimental design

The study design strikes me as reasonable and well considered.

Validity of the findings

I think this work will be of broad interest to bat and bird ecologists and others. The results appear to be well documented and presented.

Additional comments

Keep up the good work. I'll look forward to seeing this in print in the not-too-distant future. See above for specific recommendations.

---

## Round 0.2 · Minor Revisions

Thanks for thoroughly addressing the concerns of the reviewers of your original submission. Those reviewers were unavailable to review your resubmission, so I reviewed it myself. I have just a handful of comments/suggestions:

Line 110: Perhaps replace “hope” with “aim”?

Figure captions: The wording of many of the figure captions is unclear to me. For example, for the Fig. 1 caption I’m having trouble understanding what is meant by the last sentence (At each thermal distance…). For the captions on Figs. 3-5 I don’t understand exactly what was subtracted from what; what do negative vs. positive values indicate (more individuals from night vision or thermal cameras?)? Similarly, for the caption of Fig. 6 please clarify what was subtracted from what.

Line 198: is a word or two missing between “working” and “aid”?

Line 265: Please clarify how 548 was calculated. When I add the number of flying animals detected via night vision and thermal cameras using the data in table 1 (259 + 237) I get 496.

Lines 271 and 278: Insert a space between “1” and “flying”

Figs. 3-5: consider condensing these figures into a single 3 panel figure (similar to the display of Fig. 6) for ease of interpretation.

Lines 310-317: I’m having trouble seeing the patterns described here when I look at Fig. 6. Perhaps I don’t understand the Fig. 6 legend correctly. Basically, this text says that there were more detections of animals (6a), bats/non-bats (6b), and behaviors (6c) using thermal cameras closer to the towers, but the patterns in 6b seem different from 6a/6c.

Line 417: consider adding “to wildlife” after “effects” to make this sentence more specific.

---

## Round 0.3 · accepted · Accept

Thanks for addressing my feedback. Best wishes.